# Protein Intake, Metabolic Status and the Gut Microbiota in Different Ethnicities: Results from Two Independent Cohorts

**DOI:** 10.3390/nu13093159

**Published:** 2021-09-10

**Authors:** Pierre Bel Lassen, Ilias Attaye, Solia Adriouch, Mary Nicolaou, Judith Aron-Wisnewsky, Trine Nielsen, Rima Chakaroun, Emmanuelle Le Chatelier, Sofia Forslund, Eugeni Belda, Peer Bork, Fredrik Bäckhed, Michael Stumvoll, Oluf Pedersen, Hilde Herrema, Albert K. Groen, Sara-Joan Pinto-Sietsma, Aeilko H. Zwinderman, Max Nieuwdorp, Karine Clement

**Affiliations:** 1Nutrition and Obesities, Systemic Approaches (NutriOmics) Research Group, INSERM, Sorbonne Université, 75013 Paris, France; pierre.bellassen@aphp.fr (P.B.L.); solia.adriouch2@gmail.com (S.A.); judith.aron-wisnewsky@aphp.fr (J.A.-W.); 2Center for Research on Human Nutrition Ile-de-France (CRNH IdF), Nutrition Department, Assistance Publique Hôpitaux de Paris, Pitie-Salpêtrière Hospital, 75013 Paris, France; 3Department of Internal Medicine and Vascular Medicine, Amsterdam University Medical Center, Location Academic Medical Center, 1105 Amsterdam, The Netherlands; h.j.herrema@amsterdamumc.nl (H.H.); a.k.groen@amsterdamumc.nl (A.K.G.); s.j.pinto@amsterdamumc.nl (S.-J.P.-S.); m.nieuwdorp@amsterdamumc.nl (M.N.); 4Department of Experimental Vascular Medicine, Amsterdam University Medical Center, Location Academic Medical Center, 1105 Amsterdam, The Netherlands; 5Department of Public and Occupational Health, Amsterdam University Medical Center, Location Academic Medical Center, 1105 Amsterdam, The Netherlands; m.nicolaou@amsterdamumc.nl; 6Novo Nordisk Foundation Center for Basic Metabolic Research, Faculty of Health and Medical Sciences, University of Copenhagen, 1162 Copenhagen, Denmark; trine.nielsen@sund.ku.dk (T.N.); Fredrik.Backhed@wlab.gu.se (F.B.); oluf@sund.ku.dk (O.P.); 7Medical Department III–Endocrinology, Nephrology, Rheumatology, University of Leipzig Medical Center, 04109 Leipzig, Germany; Rima.Chakaroun@medizin.uni-leipzig.de (R.C.); michael.stumvoll@medizin.uni-leipzig.de (M.S.); 8MetaGenoPolis, INRAE (Institut National de Recherche Pour L’agriculture, L’alimentation et L’environnement), University Paris-Saclay, 78350 Jouy-en-Josas, France; emmanuelle.lechatelier@inrae.fr; 9Experimental and Clinical Research Center, a Cooperation of Charité-Universitätsmedizin and the Max-Delbrück Center, 10117 Berlin, Germany; Sofia.Forslund@mdc-berlin.de; 10Integrative Phenomics, 75011 Paris, France; eugenibc@gmail.com; 11Structural and Computational Biology, European Molecular Biology Laboratory, 69117 Heidelberg, Germany; bork@embl.de; 12Wallenberg Laboratory, Department of Molecular and Clinical Medicine, Sahlgrenska Academy, University of Gothenburg, 41345 Gothenburg, Sweden; 13Department of Clinical Physiology, Sahlgrenska University Hospital, 41345 Gothenburg, Sweden; 14Department of Clinical Epidemiology, Biostatistics and Bio-Informatics, Amsterdam University Medical Center, Location Academic Medical Center, 1105 Amsterdam, The Netherlands; a.h.zwinderman@amsterdamumc.nl; 15Department of Internal Medicine, Diabetes Center, Amsterdam University Medical Center, Location VU University Medical Center, 1105 Amsterdam, The Netherlands

**Keywords:** protein diet, gut microbiota, diabetes, ethnicity, HELIUS study

## Abstract

Background: Protein intake has been associated with the development of pre-diabetes (pre-T2D) and type 2 diabetes (T2D). The gut microbiota has the capacity to produce harmful metabolites derived from dietary protein. Furthermore, both the gut microbiota composition and metabolic status (e.g., insulin resistance) can be modulated by diet and ethnicity. However, to date most studies have predominantly focused on carbohydrate and fiber intake with regards to metabolic status and gut microbiota composition. Objectives: To determine the associations between dietary protein intake, gut microbiota composition, and metabolic status in different ethnicities. Methods: Separate cross-sectional analysis of two European cohorts (MetaCardis, *n* = 1759; HELIUS, *n* = 1528) including controls, patients with pre-T2D, and patients with T2D of Caucasian/non-Caucasian origin with nutritional data obtained from Food Frequency Questionnaires and gut microbiota composition. Results: In both cohorts, animal (but not plant) protein intake was associated with pre-T2D status and T2D status after adjustment for confounders. There was no significant association between protein intake (total, animal, or plant) with either gut microbiota alpha diversity or beta diversity, regardless of ethnicity. At the species level, we identified taxonomical signatures associated with animal protein intake that overlapped in both cohorts with different abundances according to metabolic status and ethnicity. Conclusions: Animal protein intake is associated with pre-T2D and T2D status but not with gut microbiota beta or alpha diversity, regardless of ethnicity. Gut microbial taxonomical signatures were identified, which could function as potential modulators in the association between dietary protein intake and metabolic status.

## 1. Introduction

Type 2 diabetes (T2D) is a major health issue which leads to high levels of morbidity and mortality on a global scale [1]. The pathophysiology of T2D is complex and it is characterized by metabolic impairment, leading to insulin resistance and eventually lower levels of systemic insulin [2].

Recently, the gut microbiota, a “new” endocrine organ, has been shown to be involved in obesity and insulin resistance [3,4]. Furthermore, the burden of T2D is not equally distributed among different ethnic groups and countries [1,5], which could possibly be explained by differences in the gut microbiota [6]. The exact relationship between T2D and the gut microbiota remains to be elucidated. However, multiple studies have shown that the gut microbiota composition and function of patients with T2D is altered compared with healthy individuals [7,8]. This alteration is mainly characterized by a decrease in diversity, which is associated with the production of deleterious gut derived metabolites, such as branch-chained amino acids (BCAA) [9], trimethylamines (TMA) [10], and imidazole propionate (ImP) [11].

Diet is one of the main modulators of metabolic status (e.g., insulin resistance), but also of gut microbiota composition and function [12,13]. Multiple studies have shown the influence of carbohydrates and dietary fibers on insulin resistance and the gut microbiota [14,15,16]. However, recent studies show that not only carbohydrates and fibers, but also dietary protein can significantly impact insulin resistance and T2D risk and progression [17,18]. Moreover, protein intake has been shown to differ according to ethnic backgrounds [19]. The source of dietary protein is also of importance, as studies show that animal protein is associated with higher cardiovascular mortality and risk of T2D compared with plant protein [17,18,20]. Furthermore, dietary protein is linked to the production of metabolites that can increase insulin resistance via the gut microbiota [21,22,23]. However, the interaction between dietary protein, gut microbiota and ethnicity with regards to metabolic status remains unclear. Therefore, the aim of this study was to investigate the interaction between dietary protein, the gut microbiota, and metabolic status, specifically focusing on insulin resistance as defined in pre-T2D and T2D subjects, in two independent cohorts from two European countries, and in different ethnicities (Caucasians vs. non-Caucasians).

## 2. Methods

### 2.1. Study Design and Populations

We examined 1759 subjects from the MetaCardis cohort and 1549 from the Healthy Life in an Urban Setting (HELIUS) cohort for whom nutritional and gut microbiota data were available.

For the MetaCardis cohort, subjects were recruited between 2013 and 2015 in clinical institutions in France (Pitié-Salpêtrière Hospital, Center of Research for Clinical Nutrition (CRNH) and Institute of Cardiometabolism And Nutrition (ICAN)), Germany (Integrated Research and Treatment Center (IFB) Adiposity Diseases in Leipzig), and Denmark (Novo Nordisk Foundation Center for Basic Metabolic Research (CBMR) in Copenhagen) for the European project MetaCardis (www.MetaCardis.net, accessed date 1 July 2021), as described previously [11]. All subjects provided written informed consent and the study was conducted in accordance with the Helsinki Declaration. The study is registered at: https://clinicaltrials.gov/show/NCT02059538. The Ethics Committee of each participating country approved the clinical investigation. The study was approved by the Comite de Protection des Personnes (CPP) Ile de France III n° IDRCB2013-A00189-36.

For HELIUS cohort, subjects were recruited between 2011 and 2015 from the municipality of Amsterdam, The Netherlands as previously described [24]. The HELIUS study was performed in accordance with the Helsinki Declaration and approved by the Institutional Research Board of the Amsterdam University Medical Centre. More information about the HELIUS study can be found on http://www.heliusstudy.nl/en/researchers/publications, accessed date 1 July 2021 [24]. 

A detailed list of prescribed medications, anthropometric data, clinical history, fecal sample, and a fasting blood sample were obtained at enrollment. Subjects were classified as healthy, pre-diabetes (pre-T2D), or type 2 diabetes (T2D). T2D was defined as fasting glycemia ≥7.0 mmol/L and/or 2 h values during the oral glucose tolerance test (OGTT) >11.1 mmol/L and/or hemoglobin A1c (HbA1c, glycated hemoglobin) ≥6.5% (≥48 mmol/mol) and/or use of any anti-diabetic treatment; pre-T2D was defined for subjects without T2D as fasting glycemia ≥5.6 mmol/L and/or 2 h values in the oral glucose tolerance test (OGTT) ≥7.8 mmol/L and/or hemoglobin A1c (HbA1c, glycated hemoglobin) ≥5.7% (≥39 mmol/mol) according to the American Diabetes Association (ADA) definitions [25,26]. 

Ethnicity was assessed according to country of birth of subject and the parents, as previously described [27].

### 2.2. Dietary Intake Data Assessment

Dietary data for the MetaCardis cohort was collected via a validated food-frequency questionnaire that was adapted to the cultural habits of each of the countries of recruitment. A validation study against repeated 24-h dietary records among 324 French MetaCardis participants indicated a good validity for macronutrients and protein food groups [28]. Portion size and nutrient composition were derived from national food consumption surveys and food composition databases. Data on physical activity were collected using a validated Recent Physical Activity Questionnaire (RPAQ) [29]. 

Dietary data for the HELIUS cohort was obtained via a validated, ethnicity-specific food-frequency questionnaire, as previously described [30].

For each subject, basal metabolic rate (BMR) was estimated using the Harris and Benedict Formula [31]. Subjects with under- or over-reporting energy intake declarations defined as <0.5*BMR or >3.5*BMR were excluded from all nutritional analysis (<10% of the subjects with available nutritional data). 

### 2.3. Biochemical Analyses

Blood samples were collected after an overnight fast. Fasting serum glucose, triglycerides, and HbA1c were measured using enzymatic methods. 

### 2.4. Extraction of Fecal Genomic DNA and Gut Microbiota Sequencing

Gut microbiota sequencing in the MetaCardis cohort was performed using shotgun sequencing as previously described [11]. Briefly, participants collected fecal samples within 24 h before each visit. Samples were either stored immediately at −80 °C or briefly conserved in home freezers before transport to the laboratory, where they were immediately frozen at −80 °C following guidelines [32]. Total fecal DNA was extracted and sequenced using ion-proton technology (ThermoFisher Scientific, WA, USA). The reads were mapped to the Integrated Gene Catalog (IGC) of 9.9 million genes [33]. Gene abundance tables (built from mapping against the 9.9M gene catalog) were processed for richness calculation, downsizing (to 10 million reads) and normalization using the momr R package.

Gut microbiota sequencing in the HELIUS cohort was performed as previously described [6]. Briefly, subjects delivered a fresh stool sample within 6 h after collection to the research location. At the research location, the samples were temporarily stored at −20 °C and transported the next day to the −80 °C freezers. The gut microbiota composition was determined using 16S rRNA, focusing on the V4 region. The RNA was sequenced using a MiSeq ststem (RTA version 1.17.28, bundled with MCS version 2.5; Illumina, San Diego, CA, USA).

### 2.5. Statistical Analysis

All nutrient data are originally expressed as grams of intake per day. In all the analysis, residuals of each nutrients to total energy intake were calculated as described previously [34] to correct for total caloric intake. Participants characteristics were analyzed using chi-square tests for categorical variables and t-tests for numerical variables, using the *TableOne* Package in R. High protein eaters were defined as protein intake >20% of total energy intake. This number is based on the fact that the average protein consumption is 14 energy percent in the Netherlands and 13 energy percent of total caloric intake in France (https://ourworldindata.org/diet-compositions, accessed date 1 July 2021). This is also in agreement with nutritional recommendations to consume a 10−20% range of total energy intake from protein [35]. The link between nutritional data and metabolic status was assessed using logistic regression with adjustment for confounders. The model used was metabolic status ~ protein_residual + carbohydrate_residual + fat_residual + fibre_intake + total_kcal_intake + age + sex + physical_activity. To investigate the links between macronutrients intake and the gut microbiota, residuals of macronutrients adjusted for energy intake and other gut microbiota confounders were computed. The model used to compute residuals was: macronutrient ~ total_kcal_intake + age + sex + Body Mass Index (BMI) + T2D status + anti diabetic treatment + lipid lowering treatment. Gut microbiota alpha diversity was assessed by computing Shannon index. Correlations between residuals of each macronutrient and alpha diversity were computed. Gut microbiota beta diversity was computed using Bray-Curtis distance in each cohort and permanova was performed to assess the contribution of each macronutrient residuals to the beta diversity using *vegan* and *adonis* R packages. The most important species as defined in (molecular) Operational Taxinomical Units (mOTUs for MetaCardis and OTUs for HELIUS) associated with animal protein intake were identified with cross-validated optimized random forest models using animal protein intake residuals adjusted for age, sex, BMI, anti-diabetic treatment, and lipid-lowering treatments with the *caret* R package. The variable importance was determined using increase in node purity. The analyses were restrained at the species OTU level.

Statistical analyses and conception of figures were carried out using R version 3.3.2, R Core Team (2019), https://www.R-project.org/, acessed date 1 July 2021.

## 3. Results

### 3.1. Cohort Characteristics

The characteristics of the two independent cohorts, MetaCardis and HELIUS, are described in Table 1. 

Briefly, the MetaCardis cohort and HELIUS cohort were similar in sex distribution (50.2% females in MetaCardis and 52.6% in HELIUS, *p* = 0.17). However, participants of the MetaCardis cohort were slightly older and had a higher percentage of individuals with T2D (37.5%) than the HELIUS cohort (16.7%), whereas the HELIUS cohort had a higher percentage of subjects with pre-T2D (57.3%) compared with the MetaCardis cohort (34.1%). The MetaCardis cohort consisted of more subjects of Caucasian ethnicity (88.4%) than the HELIUS cohort (32.4%) (*p* < 0.01). The mean protein consumption in MetaCardis was slightly higher than in HELIUS (94.4 ± 40.4 vs. 89.3 ± 35.4 g/day, *p* < 0.01) and the proportion of high protein eaters was also higher in MetaCardis (27.7% vs. 10.7%, *p* < 0.01) (for information on animal or plant protein see supplementary figures). Both cohorts from different countries were analyzed separately due to the abovementioned differences and different methods of gut microbiota sequencing. 

### 3.2. Protein Intake and Metabolic Status

Despite differences in the two cohorts, protein intake was significantly associated with the MetaCardis (pre-T2D and T2D in both MetaCardis (pre-T2D: Odds Ratio (OR) 1.14 (Confidence Interval (CI):1.05–1.24); T2D: OR 1.18 (CI: 1.08–1.29)) cohort, as well as the HELIUS (pre-T2D: OR 1.29 (CI:1.18–1.41); T2D: OR 1.73 (CI: 1.49–2.03)) cohort, after adjustment for nutritional confounders, physical activity, age, and sex (Figure 1). Interestingly, the association could almost be fully explained by animal protein intake (MetaCardis pre-T2D: 1.14 (CI: 1.05–1.24) T2D: OR 1.18 (CI: 1.08–1.29); HELIUS pre-T2D: 1.26 (CI 1.16–1.37); T2D:1 1.61 (CI: 1.40–1.87)), but not by plant protein intake (MetaCardis pre-T2D: 1.21 (CI: 0.93–1.57); T2D:1.16 (CI: 0.90–1.48) HELIUS pre-T2D: 1.09 (CI: 0.88–1.36); T2D: 1.20 (CI: 0.88–1.66), in both cohorts. Similarly, being a high protein eater, i.e., proportion of the total energy intake attributable to protein >20%, was strongly associated with T2D and pre-T2D in both cohorts (MetaCardis pre-T2D: 1.50 (CI: 1.08–2.11) T2D: 1.97 (CI: 1.40–2.77; HELIUS pre-T2D: 4.23 (CI: 2.55–7.27); T2D: 6.66 (CI: 3.40–13.56)). In MetaCardis, we observed that the association between animal protein intake and T2D status was more pronounced in Caucasians vs. non Caucasians (interaction term: 0.15, *p* = 0.027), however this was not the case in HELIUS. Furthermore, the found associations were significantly attenuated by BMI (Appendix A). This is likely due to overcorrection as BMI/obesity is involved in the causal pathway of insulin resistance [17]. We also provide univariate analysis results (Appendix A).

### 3.3. Macronutrient Intake and Gut Microbiota Alpha Diversity

We then determined the links between alpha diversity (Shannon index) and macronurient intake adjusted for energy intake, age, sex, BMI, T2D status, and lipid-lowering and antidiabetic treatment (Figure 2). We also examined potential differences depending on subjects’ ethnicity by stratifying subjects into Caucasian vs. non-Caucasian ethnicity. In both cohorts, fat intake was positively associated with alpha diversity. In MetaCardis, this association was only found regarding unsaturated fat and not saturated fat. In both cohorts, carbohydrate intake was negatively associated with alpha diversity. In contrast, protein intake showed no significant correlation with alpha diversity, regardless of the protein source. When stratified for ethnicity, the previously mentioned association between unsaturated fat and alpha diversity remained significant for subjects of both Caucasian and non-Caucasian ethnicity in both cohorts. On the other hand, the negative association observed between carbohydrates intake and alpha diversity was only observed in non-Caucasians in both cohorts with a significant interaction between ethnicity, carbohydate, and alpha diversity in MetaCardis (*p* = 0.015).

### 3.4. Macronutrient Intake and Gut Microbiota Beta Diversity

The beta diversity was calculated using the Bray-Curtis method in order to study the contribution of all macronutrients to the known explained variance of the gut microbiota (Figure 3). First, we found that the macronutrients that contributed the most to the explained variance were rather consistent in the two cohorts. Notably, dietary fiber had a significant association with beta diversity in both cohorts and was the one with the highest link in HELIUS and the one with second highest link in gut microbiota variance in MetaCardis. Interestingly, in both cohorts, total protein and animal protein had no significant association with beta diversity and were the macronutrients with the least explained variance, whereas plant protein had a significant association in both cohorts. Importantly, all these analyses were adjusted for potential confounders on microbiota composition and the significant effects are therefore independent of total caloric intake, age, sex, BMI, T2D status, and anti-diabetic and anti-lipid treatment intake (Figure 3A,B). We also stratified the analysis according to Caucasian vs. non-Caucasian ethnicity and found that total protein and animal protein were not significantly associated with beta diversity in either cohort, regardless of ethnicity (Figure 3C–F). Overall, after adjustment for confounders, protein and specifically animal protein intake were not significantly linked with gut microbial beta diversity in our two cohorts, regardless of Caucasian vs. non-Caucasian ethnicity. 

### 3.5. Species Associated with Animal Protein Intake

Since animal protein intake was linked with T2D status but was not significantly associated with gut microbiota diversity, we next sought to determine if we could nevertheless identify taxonomical signatures associated with the consumption of animal protein intake and if these species overlapped in the two independent cohorts, using random forest models adjusted for total caloric intake, age, sex, BMI, metformin, and statin intake. Among the 30 most important species associated with animal protein intake, we found 11 (37%) that overlapped between the two cohorts (Figure 4). Most of these were positively associated with animal protein intake apart from *Roseburia hominis*, which showed a negative association in both cohorts, and *Colinsella aerofaciens*, which was negatively associated with animal protein intake in the HELIUS cohort.

In the HELIUS cohort, which is more diverse in terms of different ethnicities, we found that seven of these species’ abundance were significantly associated with ethnicity (Caucasian vs. Non-Caucasian). Subjects of Caucasian ethnicity (in HELIUS) had higher abundances of *Bacteroides uniformis, Coprococcus comes, Faecalibacterium prausnitzii, Odoribacter splanchnicus*, and *Parabacteroides distasonis* but significantly lower levels of *Prevotella copri* and *Dorea formicigenerans* compared with non-Caucasians. However, we did not find the same significance in MetaCardis, where the majority of the population consisted of Caucasians (Appendix A). For some of these species, such as *Bacteroides uniformis* or *Parabacteroides distasonis,* we found a positive association with T2D status in MetaCardis; but it was not confirmed in HELIUS (Appendix A). On the other hand, *Roseburia hominis,* which was negatively associated with animal protein, was decreased in T2D patients in both cohorts.

## 4. Discussion

In this cross-sectional study design, we showed, for the first time, the interaction between protein intake, metabolic status, and gut microbiota composition in different ethnic groups (Caucasian vs. non-Caucasian) in two large, independent cohorts from different European countries. Previous research has mainly focused on the role of fiber intake with regards to metabolic status and the gut microbiota, whereas recent studies highlight the importance of protein intake and ethnicity [6,13]. This study observed an association between animal protein intake and a worsened metabolic status, confirming previous findings in different populations [17,36]. We did not observe significant associations between animal protein intake and gut microbiota alpha or beta diversity, regardless of ethnicity. However, we reported several species associated with animal protein intake in both cohorts, which could serve as potential targets for future studies. Surprisingly, we found that fat intake was positively associated with microbial alpha diversity in both cohorts. Nevertheless, when stratifying these analyses for fat source, we found that this association is mostly attributable to unsaturated fat. However, a recent clinical trial did not find any effects of a high fat diet on alpha diversity [37], which might be explained by differences in fecal matter collection and sequencing techniques, as well as with various food intake patterns in different studied populations.

This study is in line with the findings of a previous large meta-analysis, which showed that high consumption of total protein and animal protein is associated with T2D with a relative risk of 1.12 and 1.14 respectively [38]. In our studied populations, protein intake was associated with T2D status. This association had an odds ratio of 1.19 in the MetaCardis cohort and 1.73 in the HELIUS cohort and was largely explained by the consumption of animal protein in both cohorts. In MetaCardis, we observed a significant interaction between animal protein intake, T2D status, and ethnicity, i.e., the association between animal protein intake was overall stronger in Caucasians than in non-Caucasians.

The precise biological mechanism via which (animal) protein intake could increase T2D risk remains unknown. However, recent evidence points towards gut microbial produced metabolites as potential mediators between diet and metabolic status [13]. More specifically, the increased production of harmful metabolites which derive from protein degradation such as Branched chain amino acids (BCAA), Imidazole Propionate (ImP), or Trimethylamine N-oxide (TMAO) have been linked with the development of metabolic diseases and their cardiovascular complications [11,23,39]. We therefore investigated whether this association could be mediated by the gut microbiota. However, in both cohorts, we found that animal protein, after adjustment for confounders, was not correlated with gut microbial diversity and that among all macronutrients, it was the one with the least effect on beta diversity. Importantly, macronutrients only explained a small percentage of the beta diversity, which is in line with previous studies [40,41]. It has been suggested that there are more unknown covariates such as intrinsic microbial ecological processes, species interactions, and dynamics which could influence microbial diversity to a larger degree then nutrition [41], which is currently recognized as one of the major modifiable factors in gut microbiota composition and function [13]. Since a previous study showed that ethnicity strongly impacts gut microbiota composition, regardless of diet, and plays a larger role than traditionally recognized factors such as obesity and BMI [42], we next investigated whether stratification by ethnicity would change our findings. However, this was not the case in either cohort. 

This study next sought out to investigate if specific microbial species were associated with animal protein intake. Despite different methodological methods in the gut microbiota assessment techniques between MetaCardis and HELIUS, we observed 11 similar species shared in both cohorts among the 30 most important strains that explained animal protein intake. Of the 11 shared species, *Roseburia hominis* was the only one that was negatively associated with animal protein intake in both cohorts. This bacterial strain has not been studied in a dietary setting. However, *Roseburia hominis* has been identified as a major butyrate producer [43], which is associated with improved metabolic status [44]. The closely associated *Roseburia inulivorans*, on the other hand, was positively associated with animal protein intake in both cohorts. This finding highlights the diverse effects of protein intake on (closely related) bacterial species, warranting further studies in order to modulate metabolic status. 

This study also identified previously described species that were negatively associated with metabolic status [45,46,47,48,49]. A previous study has shown that adherence to a Mediterranean Diet negatively impacted the abundance of several species which were here positively associated with animal protein intake [49]. These species were *Collinsella aerofaciens, Coprococcus comes,* and *Dorea formicigenerans*. Of these species, *Collinsella aerofaciens* was shown to be highly abundant in patients with metabolic syndrome and obesity and showed a strong positive correlation with high triglycerides, and low HDL. This bacterial strain is suggested to be a biomarker for obesity and metabolic syndrome and has also been associated with low fiber intake in a previous study [50,51]. Another species which was identified in our study was *Bilophila wadsworthia,* a bile-tolerant gram-negative rod. This bacteria is increased with animal-based diets in humans and is recognized as a pathobiont, which can cause metabolic derangements when exposed to a diet rich in lipids, as shown in HFD mice [47,48]. 

Moreover, eight of the identified bacterial species associated with animal protein intake had different abundances depending on ethnicities, the majority of them being more abundant in Caucasians vs. non-Caucasians. This increase in animal protein-associated bacteria could perhaps contribute to the more pronounced association of animal protein intake observed in Caucasians vs. non-Caucasians. This finding again highlights the importance of an ethnic specific approach to gut microbiota research [52].

Several limitations exist in this study. First, the MetaCardis and HELIUS cohorts differ in their population composition, which made it impossible to pool the data from both cohorts. Second, both cohorts used different methods (i.e., shot gun metagenomics vs. 16S rRNA sequencing) to characterize gut microbiota composition and function. We were therefore limited in exploring full functional pathways in both cohorts. Third, the nature of this study was cross-sectional, making it prone to reverse causality (i.e., T2D subjects can alter their nutritional intake towards more protein intake which can also influence gut microbiota composition).

However, the fact that this study found similar associations between (animal) protein intake, metabolic status, gut microbiota composition, and ethnicity in both cohorts, despite the abovementioned differences, leads to greater generalizability of the results in different population settings. 

## 5. Conclusions

In this study, we confirmed that dietary (animal) protein is associated with poor metabolic status, but is not associated with the gut microbiota composition and diversity regardless of ethnicity (Caucasian vs. non-Caucasian), taking into account confounders. We nevertheless identified several species linked with animal protein intake that could serve as targets for future intervention studies. Moreover, this study identified ethnicity as a modifier in the interaction between diet, metabolic status, and gut microbiota composition. Future studies are needed with well-characterized ethnic groups, in-depth microbiota sequencing techniques, and detailed dietary data in order to further shed light on the complex interplay between diet, health, and the gut microbiota.

## Figures and Tables

**Figure 1 nutrients-13-03159-f001:**
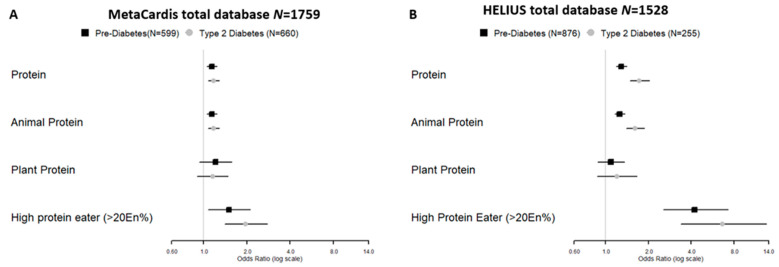
Forrest plots representing protein intake in the (**A**) MetaCardis and (**B**) HELIUS cohort. Data is shown per 10 g of protein intake and the model was adjusted for age, gender, physical activity, other macronutrients and total energy intake according to the energy residual model. Model used: Metabolic status = protein_residual + carbohydrate_residual + fat_residual + fibre_intake + total_kcal_intake + age + gender + physical_activity.

**Figure 2 nutrients-13-03159-f002:**
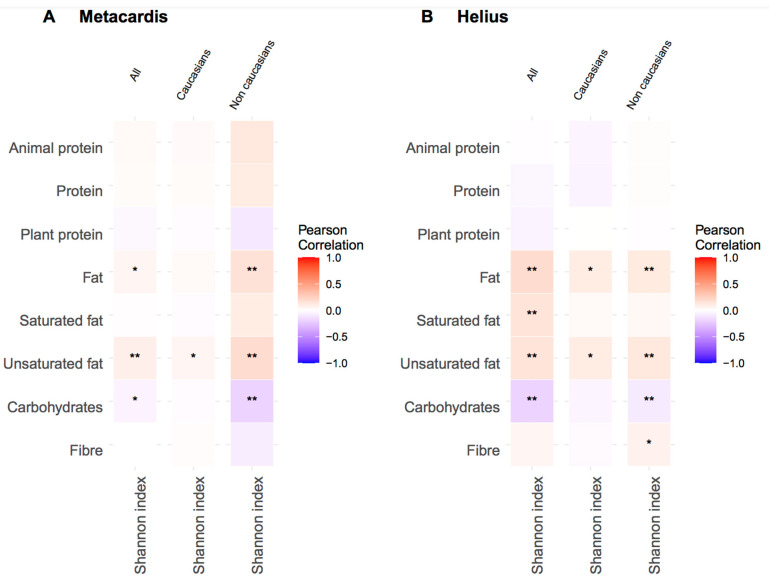
Association between macronutrient intake and alpha-diversity (Shannon Index) stratified on ethnicity in (**A**) MetaCardis cohort and (**B**) HELIUS cohort. Macronutrients are adjusted on energy intake, age, gender, BMI, T2D status, anti diabetic treatment and anti lipidic treatment using residuals. * *p <* 0.05; ** FDR adjusted *p <* 0.05.

**Figure 3 nutrients-13-03159-f003:**
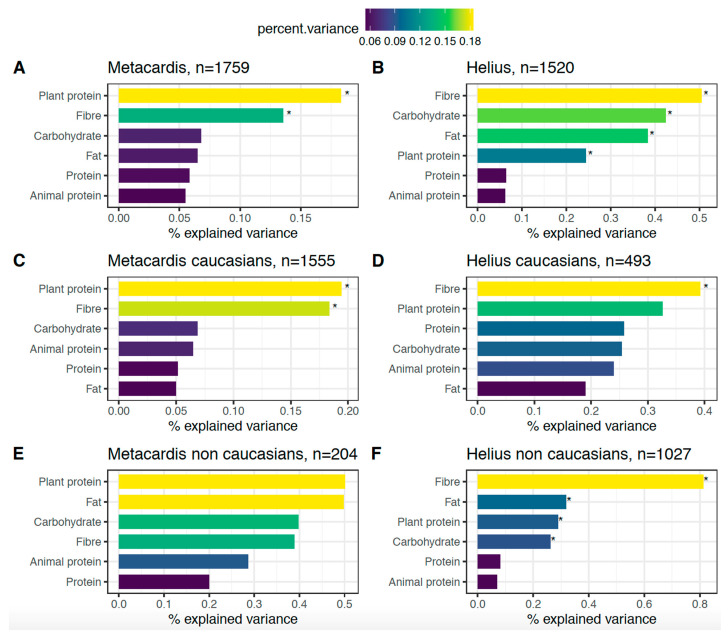
Association between macronutrient intake and beta diversity in the (**A**,**C**,**E**) MetaCardis cohort and (**B**,**D**,**F**) HELIUS cohort. Macronutrients are adjusted on energy intake, age, sex, BMI, T2D status, anti-diabetic treatment and anti-lipidic treatment using residuals. * *p* < 0.05.

**Figure 4 nutrients-13-03159-f004:**
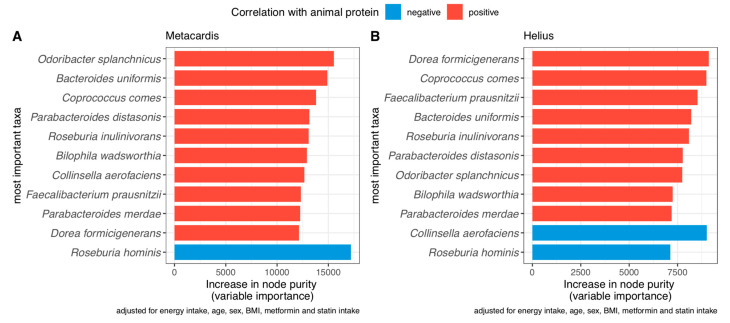
Most important taxa associated with animal protein intake with overlap in both cohorts (**A**) MetaCardis, (**B**) HELIUS.

**Table 1 nutrients-13-03159-t001:** Baseline characteristics of the MetaCardis (A) and HELIUS (B) cohort.

	All	Control	Pre-T2D	T2D
**(A) MetaCardis**				
	100% (*N* = 1759)	28.4% (*N* = 500)	34.1% (*N* = 599)	37.5% (*N* = 660)
Age (years)	57.0 ± 12.0	51.7 ± 14.2	57.2 ± 11.1 *	60.8 ± 9.1 *
Sex: Female (%)	50.2 (*N* = 883)	61.4% (*N* = 307)	46.2 (*N* = 277) *	45.3 (*N* = 299) *
Ethnicity:				
Caucasians (%)	88.4 (*N* = 1555)	93.4 (*N* = 467)	91.3 (*N* = 547)	82.0 (*N* = 541)
Non Caucasians (%)	11.6 (*N* = 204)	6.6 (*N* = 33)	8.7 (*N* = 52)	18.0 (*N* = 119) *
BMI (kg/m^2^)	32.1 ± 8.7	30.1 ± 9.8	32.4 ± 8.9 *	33.3 ± 7.1 *
Total physical activity (MET/h/week)	96.3 ± 75.1	104.4 ± 74.3	93.7 ± 72.5	92.5 ± 77.5 *
Diabetes treatment (%)	30.6 (*N* = 539)	0.0 (*N* = 0)	0.0 (*N* = 0)	81.7 (*N* = 539) *
Statin treatment (%)	35.5 (*N* = 624)	21.0 (*N* = 105)	33.7 (*N* = 202) *	48.0 (*N* = 317) *
Energy (kcal/day)	2128.3 ± 841.4	2134.4 ± 846.7	2189.7 ± 883.6	2068.0 ± 793.7 *
Protein (g/day)	94.4 ± 40.4	92.3 ± 38.2	96.8 ± 44.1	93.7 ± 38.4
Protein (% of total energy intake)	18.4 ± 3.4	18.0 ± 3.5	18.4 ± 3.3	18.7 ± 3.2 *
Animal protein (g/day)	62.3 ± 33.4	60.0 ± 31.4	64.4 ± 37.3	62.1 ± 31.0
Animal protein (% of total energy intake)	11.9 ± 4.0	11.5 ± 4.2	11.9 ± 3.9	12.1 ± 3.8 *
Plant protein (g/day)	32.1 ± 14.5	32.3 ± 14.7	32.4 ± 14.6	31.6 ± 14.2
Plant protein (% of total energy intake)	6.1 ± 1.6	6.1 ± 1.5	6.0 ± 1.6	6.2 ± 1.5
Fat (g/day)	77.6 ± 34.7	78.3 ± 33.8	80.1 ± 37.1	74.8 ± 32.9
Carbohydrates (g/day)	249.0 ± 109.6	251.1 ± 115.4	256.5 ± 108.8	240.6 ± 105.2
Fiber (g/day)	29.5 ± 15.1	29.8 ± 15.2	29.2 ± 14.5	29.6 ± 15.5
High protein (>20En%) eater (%)	27.7 (*N* = 488)	22.6 (*N* = 113)	26.5 (*N* = 159)	32.7 (*N* = 216) *
**(B) HELIUS**				
	100% (*N* = 1528)	29.3% (*N* = 447)	57.3% (*N* = 876)	16.7% (*N* = 255)
Age (years)	52.20 ± 10.55	46.14 ± 11.97	54.45 ± 8.70 *	57.97 ± 7.36 *
Sex: Female (%)	52.6 (*N* = 804)	61.7 (*N* = 276)	48.9 (*N* = 428) *	50.6 (*N* = 129) *
Ethnicity:				
Caucasians (%)	32.4 (*N* = 495)	43.4 (*N* = 194)	30.0 (*N* = 263)	16.9 (*N* = 43)
Non Caucasians (%)	67.6 (*N* = 1033)	56.6 (*N* = 253)	70.0 (*N* = 613)	83.1 (*N* = 212)
BMI (kg/m^2^)	27.15 ± 4.76	24.98 ± 4.03	27.69 ± 4.60 *	29.50 ± 5.00 *
Total physical activity (h/week)	43.82 ± 28.35	45.57 ± 25.32	44.43 ± 30.35	40.12 ± 31.36 *
Diabetes treatment (%)	10.3 (*N* = 157)	0.0 (*N* = 0)	6.7 (*N* = 59) *	61.6 (*N* = 157) *
Statin treatment (%)	14.0 (*N* = 218)	7.6 (*N* = 99)	14.3 (*N* = 125) *	46.7 (*N* = 119) *
Energy (kcal/day)	2226.11 ± 822.92	2253.29 ± 764.94	2237.58 ± 854.61	2155.57 ± 875.51
Protein (g/day)	89.26 ± 35.41	87.50 ± 31.72	90.33 ± 37.10	90.68 ± 40.02
Protein (% of total energy intake)	16.2 ± 3.2	15.7 ± 3.0	16.3 ± 3.3 *	17.0 ± 3.3 *
Animal protein (g/day)	52.26 ± 27.01	50.11 ± 24.29	53.46 ± 28.18 *	54.00 ± 31.13
Animal protein (% of total energy intake)	9.5 ± 3.6	9.0 ± 3.3	9.7 ± 3.7 *	10.0 ± 3.8 *
Plant protein (g/day)	36.99 ± 15.31	37.38 ± 14.77	36.87 ± 15.77	36.68 ± 14.93
Plant protein (% of total energy intake)	6.7 ± 1.6	6.7 ± 1.7	6.7 ± 1.6	7.0 ± 1.7 *
Fat (g/day)	79.35 ± 35.81	81.28 ± 34.69	79.42 ± 36.40	76.31 ± 36.91
Carbohydrates (g/day)	249.63 ± 104.98	251.28 ± 96.03	251.43 ± 111.04	241.89 ± 106.64
Fiber (g/day)	24.11 ± 9.47	24.55 ± 9.46	23.97 ± 9.65	24.16 ± 9.20
High protein (>20En%) eater (%)	10.7 (*N* = 163)	5.1 (*N* = 23)	13.2 (*N* = 116) *	15.7 (*N* = 40) *

Data are represented as *N* (%) for categorical data and mean (SD) for continuous data. Significance was calculated comparing to the control group. *: *p* < 0.05 for comparison against control group. Student *t* test performed for continuous data and Chi-square test performed for categorical data. MET: Metabolic equivalent of task.

## Data Availability

The data presented in this study are available on request from the corresponding author. The data are not publicly available due to ethic reasons.

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
