# Peer review of "Protein Intake, Metabolic Status and the Gut Microbiota in Different Ethnicities: Results from Two Independent Cohorts"

_nutrients, 2021, doi:10.3390/nu13093159_

Round 1

Reviewer 1 Report

[Nutrients] Manuscript ID: nutrients-1345363

Major comments

Page11 Line292, there is no discussion.

Comparing the data in Table 1 with the graph in Figure 1, there were cases where no significant difference seemed likely, but were assumed to be significant, and vice versa. (In the case of Metacardis High protein and HELIUS protein). Please re-check and analyze those data.

Minor comments

Page4 Line169, insert “degree (Ëš)” between “20 C” and “-80 C”.

Page5, Line202 Table1:

I found some of the authors’ explanations difficult to follow about the Table 1 data. In Table 1, “Data is represented as mean±SD and significance was calculated compare to the control group.”; however, values are seem to show mean value with parenthesis. Moreover, some case, a value in parenthesis represents SD, another case, displays percentile. Authors should show how significant difference between control and treatment group. Asterisk means so what? I suggest authors should improve explanation and display of the data in Table 1. The word “data” is a plural form.

Page6, Line 217 Figure 1:

Align the units on the x-axis in HELIUS. Bars display over range of x-axis. X-axis font is too small, it is better increase font size more larger in Metacardis and HELIUS. Both group, y-axis lines and pre-diabates group lines visual information are too weak. More thick, dark color is preferred. 

Page6, Line220, I assume “OR” means odds ratio, the first time, please spell out.

fig 3d fat

Figure S1A,  a legend, caucasian means what?

Reviewer 2 Report

Thank you for the opportunity to review this manuscript. The authors have investigated the association between protein intake, metabolic status, and the gut microbiota based on two independent cohorts. There are a few concerns for the authors to consider.

My major concerns:

  1. Discussion is missed out in the manuscript. I guess the authors forgot to copy this section in the Nutrients submission template.
  2. As this is a cross-sectional analysis, it is hardly to identify if diabetes influences protein intake and gut microbiota or vice versa. The authors may need to discuss about this.
  3. For Figure 1, it may not be sufficient to just report the adjusted odds ratios. It is better to present the unadjusted odds ratios and the number of events and controls for high protein eaters and non-high protein eaters. The analysis for high protein eaters needs to be explained in the footnotes.
  4. For Figure 3, it is so small if the numbers of variance explained are in percentages.

Minor concerns:

  1. Page 5, Title of Table 1: data is represented as mean ± SD. I cannot find this kind of figures in the Table.
  2. There are so many unnormal marks in the article. For example, Table 1: Non_Caucasians, Total_physical_activity; Figure 1: Pre_diabetes. The authors may need to check throughout the manuscript.
  3. The description for adjustment is repeated in Figure 4.
  4. Why BMI is not adjusted for in Figure 1?
  5. Why high protein eaters were defined as protein intake>20% of total energy intake? Is it arbitrary?

Round 2

Reviewer 1 Report

The study is interesting, however, manuscript style and presentation need revise. Please refer to the attached file.

Reviewer 2 Report

Thanks. The authors have well addressed my comments. I have no further comments.